# The singularity response reveals entrainment properties of the plant circadian clock

Kosaku Masuda [1,2], Isao T. Tokuda [3], Norihito Nakamichi[4] & Hirokazu Fukuda [1✉]

Circadian clocks allow organisms to synchronize their physiological processes to diurnal variations. A phase response curve allows researchers to understand clock entrainment by revealing how signals adjust clock genes differently according to the phase in which they are applied. Comprehensively investigating these curves is difficult, however, because of the cost of measuring them experimentally. Here we demonstrate that fundamental properties of the curve are recoverable from the singularity response, which is easily measured by applying a single stimulus to a cellular network in a desynchronized state (i.e. singularity). We show that the singularity response of *Arabidopsis* to light/dark and temperature stimuli depends on the properties of the phase response curve for these stimuli. The measured singularity responses not only allow the curves to be precisely reconstructed but also reveal organ-specific properties of the plant circadian clock. The method is not only simple and accurate, but also general and applicable to other coupled oscillator systems as long as the oscillators can be desynchronized. This simplified method may allow the entrainment properties of the circadian clock of both plants and other species in nature.

[1] Graduate School of Engineering, Osaka Prefecture University, Osaka, Japan. [2] Research Fellow of Japan Society for the Promotion of Science, Tokyo, Japan. [3] Graduate School of Science and Engineering, Ritsumeikan University, Shiga, Japan. [4] Division of Biological Science, Graduate School of Science, and Institute of Transformative Bio-Molecules, Nagoya University, Nagoya, Japan. ✉email: fukuda@me.osakafu-u.ac.jp

M ost organisms have circadian clocks to adapt to diurnal environmental cycles[1]. In plants, the circadian rhythms of many physiological processes respond to zeitgeber stimuli, whose types and strengths vary among organs[2]. Through sequential responses to various zeitgeber stimuli, the circadian rhythm is entrained to cyclic environmental signals with a specific locking phase. The entrained circadian phase coordinates suitable activation timing of each physiological process[3]. In the theory of nonlinear dynamics, such entrainment is primarily determined via the phase response curve (PRC)[4]. The PRC $g(\phi)$ describes dependence of the phase shift induced by external stimuli on the injection phase $\phi$. Key features of the PRC, such as its magnitude (response strength) and stable phase point (locking phase), determine the system's transient process and entrainment stability. For comprehensive understanding and control of the plant circadian clock, the PRC should be precisely measured. In the conventional method, phase shifts are measured by applying external stimuli at various circadian phases[5]; this requires many individual plants and lengthy experiments, as only a single perturbation can be applied to each plant. Moreover, the PRC may depend on clock genes[6], organs/tissues[7,8], and oscillation amplitude[9]. A less tedious method using fewer plants is needed.

We focus on a singularity state of the plant circadian clock to measure PRC simply. Singularity implies that the cellular clocks constituting the whole plant have become desynchronized from each other, diminishing the circadian amplitude of the averaged gene expressions[10]. Because the desynchronized cellular phases are widely distributed, a single external stimulus suffices to elicit a variety of phase-dependent properties of the circadian cells simultaneously. We call this single-stimulus response of the desynchronized clock system a singularity response (SR). We show as its main advantage that the PRC can be recovered from the SR.

## Results

**Construction of PRCs from singularity response.** The SR can be quantified by the peak phase $\Theta'$ and amplitude $R'$ of the circadian oscillation after the single stimulus. Figure 1 shows an example of the SR observed in *Arabidopsis thaliana* using a reporter luciferase gene (*LUC*)[11]. First, normalized bioluminescence of the rhythm of *CIRCADIAN CLOCK-ASSOCIATED 1* (*CCA1*) expression, which peaks in the morning, exhibited dumped oscillations under continuous light (LL). After almost all individual plants converged to a singularity state over 90% amplitude reduction, it responded to a 2-h dark pulse (Fig. 1a). Although the circadian rhythms of individual plants were initially damped and desynchronized with each other, they were reset to almost the same phase after the pulse (Fig. 1b). The SR, therefore, does not depend upon the state before the pulse injection. By taking the average over individuals, the synchronized phase $\Theta'$ and amplitude $R'$ are obtained after the pulse stimulus (Fig. 1c). As shown later, the SR is easily measured for a variety of zeitgeber stimuli, organs/tissues, and genotypes (Supplementary Fig. 1).

Next, we show that the PRC can be constructed from the SR. Our theory is based on the phase oscillator model

$$\frac{d\phi_j}{dt} = \omega_j + z\left(\phi_j\right)E(t).\quad(1)$$

$\phi_j$ represents the phase of the *j*th cell ($j = 1, 2, ..., N$), $\omega_j$ the intrinsic angular frequency ($\omega_j = 2\pi/\tau_j$, $\tau_j = 23$ h in this case), $z(\phi_j)$ the phase sensitivity function, and $E(t)$ the environmental input. This model has been widely applied in science and engineering to describe coupled oscillator systems including circadian oscillators[12,13]. We assume that the cellular interactions are weak and can be ignored[4,8,14], then we take only the first-

order Fourier component: $z(\phi_j) = a \sin(\phi_j - \alpha)$, parameterized by $a$ and $\alpha$. By numerical calculation, the SR ($\Theta'$, $R'$) associated with $a$ and $\alpha$ can be computed in advance. Given experimental SR data with $\Theta'$ and $R'$, $a$ and $\alpha$ can be inversely estimated using the optimization method[15,16] (Supplementary Fig. 2). Then, by calculating the model, the PRC can be drawn. To improve the estimation accuracy, a calibration is also applied (Supplementary Fig. 3).

Figure 2 shows PRCs constructed from the SR. The PRCs for 2-h dark pulse, cooling and heating (4 h, ±10 °C), and additional blue-light stimuli were measured for a *CCA1::LUC* plant (Fig. 2a–d). The red lines are the PRCs $g(\phi)$ constructed from the SR; the dots are those measured using the standard method[4,9,17]. To refer to the environmental responses in nature, circadian time (CT) was used as a timescale ("Methods" section). Most PRCs can be categorized as type-1 (showing relatively small response magnitudes). The constructions capture essential features such as the zero-crossing points and the magnitudes of the experimentally measured PRCs. Similar results are obtained for another clock gene, *TIMING OF CAB EXPRESSION 1* (*TOC1*; Supplementary Fig. 4). Although the PRC for 4 h darkness is inclined, this asymmetric shape is well described by the constructed PRC (Fig. 2e). Figure 2f–h shows PRCs for 2 h darkness in plants with mutation in *PSEUDO-RESPONSE REGULATORS 5*, *7*, or *9* (*prr5*, *prr7*, or *prr9*); *prr5* and *prr9* are slightly short and long period mutants, respectively[11]. The larger magnitude for *prr7* (Fig. 2g) implies that its oscillation is very sensitive to external stimuli, producing a type-0 PRC (having a breakpoint), and that *PRR7* plays a role in maintaining CT against light changes[18]. Thus, our SR technique is applicable also to clock mutants.

**Singularity response for various light and temperature stimuli.** From the precisely constructed PRCs, we see that the SR contains rich information on the phase response of the plant circadian clock. The SR can be measured with various light and temperature stimuli (Supplementary Fig. 1), and does not depend on stimulus timing (Supplementary Fig. 5). Moreover, the SR has been observed in *TOC1*, as well as in clock mutants (Supplementary Fig. 1d). This suggests that the SR provides a versatile, simple tool for characterizing the plant circadian clock. To address physiological responses and their environmental adaptability, SRs were measured for 11 other stimuli (Supplementary Table 1).

Figure 3 displays the extracted $\Theta'$ and $R'$ circumferentially and radially, respectively. In *CCA1::LUC*, $\Theta'$ was around midnight (CT 20–21), noon (CT5–CT7), nighttime (CT16–CT21), and late noon (CT7–CT9) for dark, heating, cooling, and blue-light stimuli, respectively, showing how the circadian clock responds to different stimuli with specific reset phases. The observed reset phases correspond well with environmental responses in nature[2]. *TOC1*, whose expression peaks in the evening, responded similarly to *CCA1::LUC* except with long-darkness stimuli. As the dark pulse duration increased from 4 to 8 h, $\Theta'$ advanced by 0.6 h in *CCA1::LUC* but by 1.4 h in *TOC1::LUC*. The difference between these two clock genes is consistent with previous studies[6] and also appeared in light–dark cycle entrainment (Supplementary Fig. 6).

With heating stimuli, temperature change influences $R'$ more strongly than stimulus duration. As the temperature change was increased from +3 °C to +10 °C (4-h heating), $R'$ became ~4 times larger, and $\Theta'$ was slightly advanced by 0.4 h (Fig. 3a), but as the heating (+3 °C) duration was lengthened from 2 to 4 h, $R'$ and $\Theta'$ remained nearly unchanged. Cooling stimuli, in contrast, showed only a minor effect. A decrease in temperature change

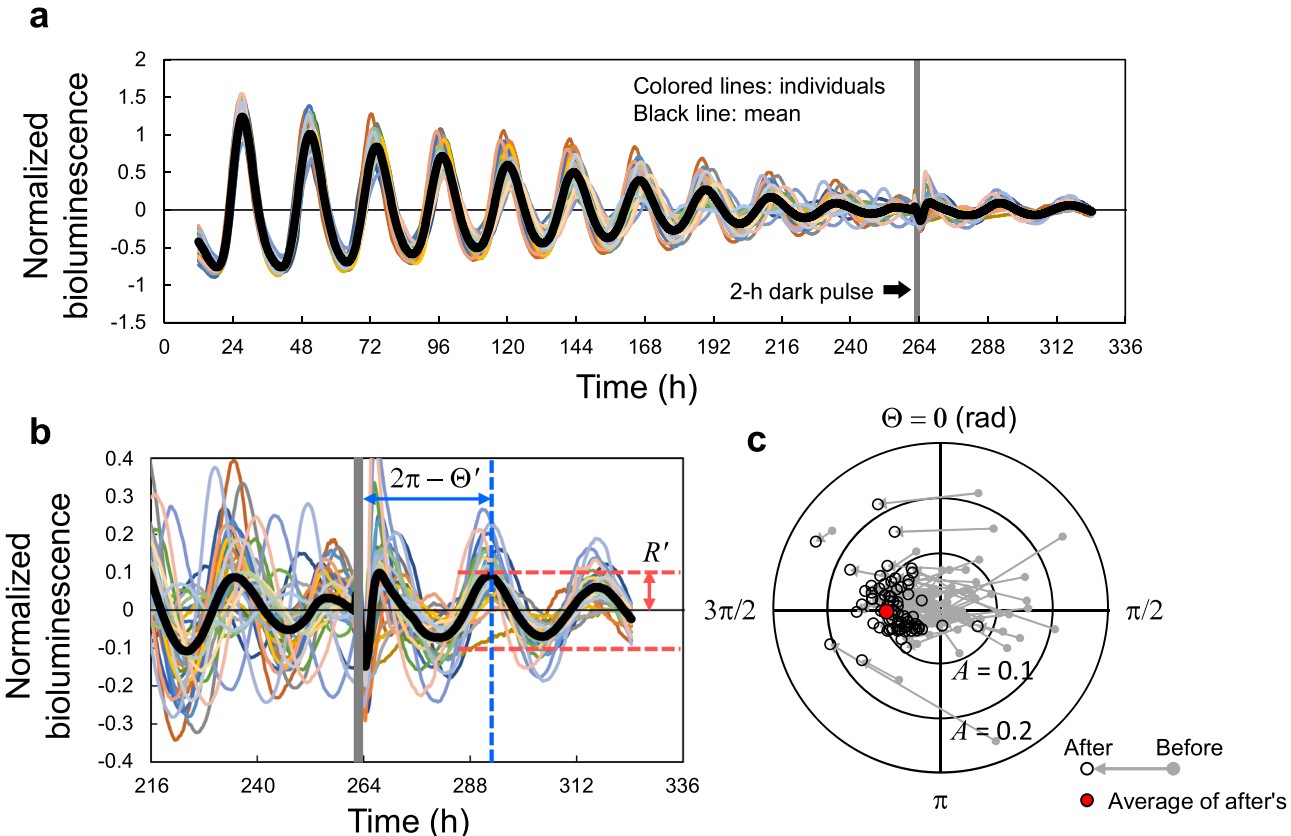

**Fig. 1 Reset of circadian rhythm at singularity state. a, b** Normalized bioluminescence of *CCA1::LUC* in continuous light (*n* = 36 individuals). The black line represents mean bioluminescence; the gray bar represents the single 2-h dark pulse. **c** Phase Θ and amplitude *A* before (gray dot) and after (black circle) pulse. Each arrow represents an individual. The red dot indicates the center of gravity of points after the pulse.

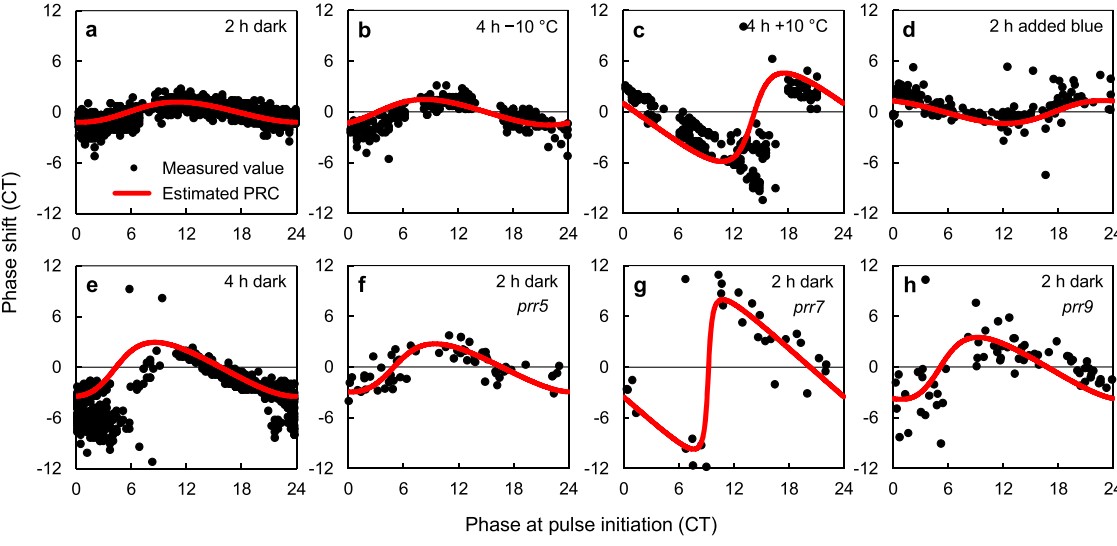

**Fig. 2 Construction of PRC from SR. a–e** PRCs for 2-h dark (**a**), 4-h 10 °C cooling (**b**), 4-h 10 °C heating (**c**), additional 2-h blue-light (**d**), and 4-h dark (**e**) stimuli measured for *CCA1::LUC* plant. **f–h** PRCs for 2-h darkness in clock mutants (*prr5* (**f**), *prr7* (**g**), and *prr9* (**h**)) in *CCA1::LUC*. Red lines are PRCs constructed from the SR.

from −3 °C to −10 °C (4-h cooling) produced only a small change in both *R′* and Θ′. As the cooling (−3 °C) duration was increased from 2 to 4 h, *R′* remained nearly unchanged but Θ′ was advanced by a few hours.

In general, physiological processes differ in their response to heating and cooling. Therefore, our results suggest that different entrainment processes should be observed for heating and cooling

stimuli. Different clock genes may also play distinct roles in entrainment to diurnal environmental changes[19].

**Organ specificity in phase response for thermal stimuli.** Finally, we investigated organ specificity as another key for describing the diversity of environmental responses in plants[7,8], focusing on shoots and roots. We separated shoot

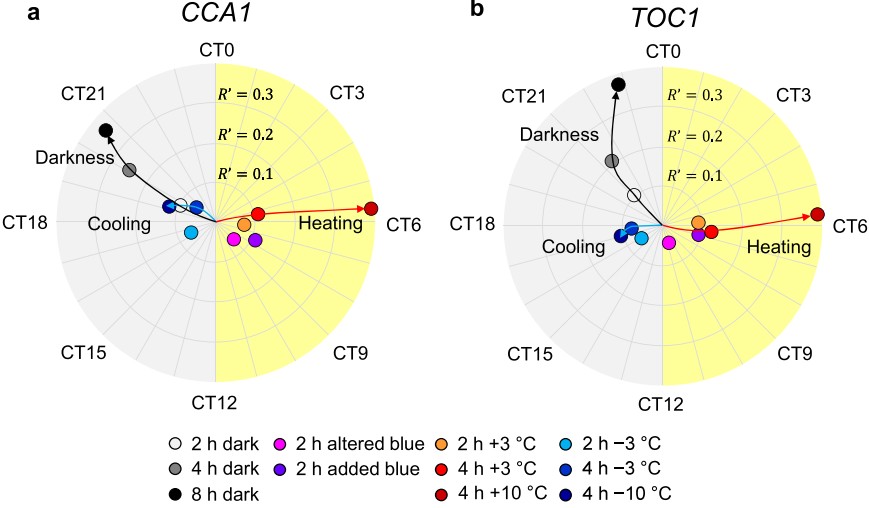

**Fig. 3 Mapping and comprehensive comparison of SRs for various stimuli. a, b** Circadian time (CT) of $\Theta'$ and $R'$ in *CCA1::LUC* (**a**) and *TOC1::LUC* (**b**). $R'$ is the distance from the center point. The graphs represent the results as means ($n = 17$–$72$).

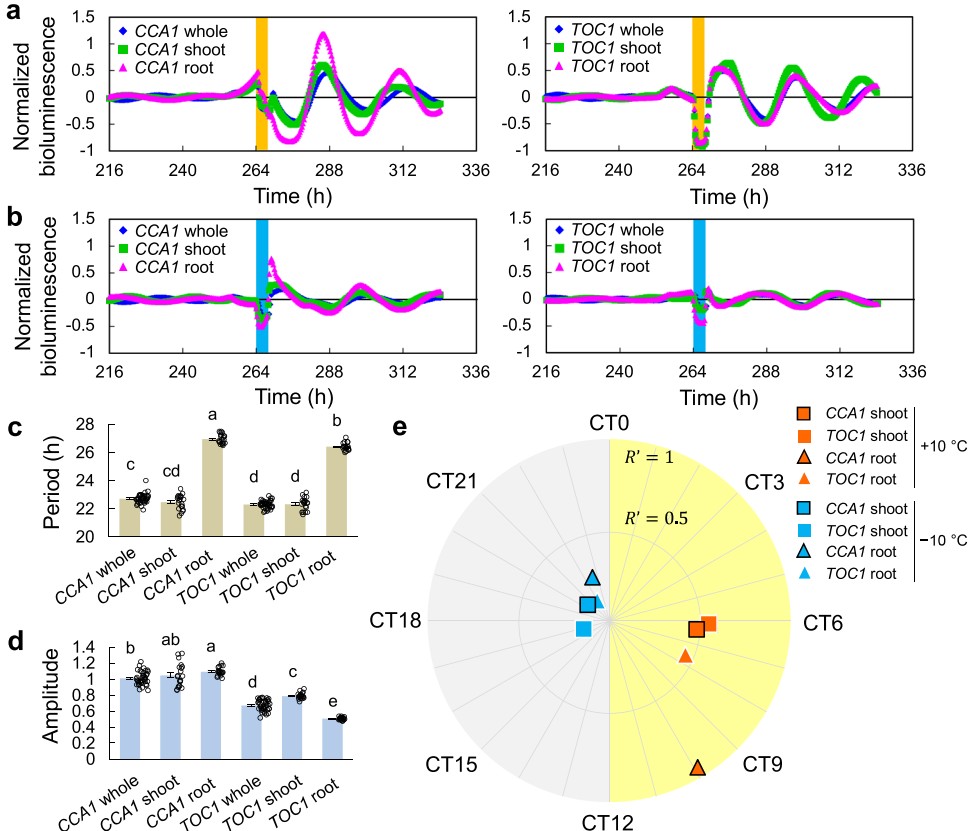

**Fig. 4 Organ specificity in SR. a** Reset of circadian rhythm against $+10\,°C$ stimulus for 4 h ($n = 20$ in whole plant and 10 in organs, mean ± standard error (SEM)). **b** Reset of circadian rhythm against $-10\,°C$ stimulus for 4 h ($n = 20$ in whole plant and 10 in organs, mean ± SEM). **c** Intrinsic periods ($n = 40$ in whole plant and 20 in organs, mean ± SEM). **d** Amplitude at beginning of measurement ($n = 40$ in whole plant and 20 in organs, mean ± SEM). In **c** and **d**, two values which have no common letter indicate significant differences (Tukey–Kramer test, $p < 0.05$). The circles indicate the individual data. **e** CT mapping of $\Theta'$ and $R'$. The graph represents the results as means ($n = 10$).

and root after 13 days' growth and measured their responses against heating and cooling stimuli (4 h, $\pm10\,°C$) in the same way as in the whole-plant experiments. Figure 4a, b shows SRs measured for shoot, root, and whole plant. The shoot and root SRs were similar to that of the whole plant, although there were large differences in their oscillation properties: the intrinsic periods (free-running period in LL) in the root were longer

than those for the shoot and whole plant (Fig. 4c), and the intrinsic oscillation amplitudes of the *CCA1* root, measured on day 1 after LL, were larger than that for the whole plant. Amplitudes for the *TOC1* root, however, were smaller than for the others (Fig. 4d). These differences in oscillation features between *CCA1* and *TOC1* are consistent with the results of previous studies[20].

Between heating ($+10\,^{\circ}\text{C}$) and cooling ($-10\,^{\circ}\text{C}$) stimuli, opposite phases $\Theta'$ were observed (Fig. 4e). Noteworthy is the large $R'$ observed for *CCA1* root in response to the heating stimulus, in contrast to that of *TOC1* root, indicating that the entrainment properties are gene specific, as well as organ specific. In some studies, the root circadian clock is considered a peripheral of the shoot's[21]. Organ specificity has also appeared in differences in amplitude response to light stimuli[20].

## Discussion

In summary, we have introduced the concept of SR to the study of plant circadian clocks. Our experiments showed that the SR can be measured for a variety of stimuli, clock genes, tissues/organs, and genotypes. The advantage of the SR is its capability to recover the PRC. The SR technique requires considerably less experimental resource than the standard method for measuring the PRC, since a single stimulus suffices to unveil phase-dependent properties of the circadian system. Indeed, the PRCs recovered from the SRs captured essential features of various PRCs reported in the previous studies[9,22–25].

To precisely recover the PRC, the SR should be measured ideally from a plant in which cellular oscillators are completely desynchronized, and thus oscillation amplitude of the bioluminescence signal is zero. It was, however, hard to realize such a zero-amplitude state experimentally. As shown in Supplementary Fig. 1, slight levels of amplitudes remained in individual plants at the timing of stimulus. To reduce the effect of such incomplete desynchronization, we averaged the SR quantities over individual plants so that the oscillation amplitude can be considerably reduced and the result can be regarded as a SR of a large population of cells (Supplementary Fig. 5).

In our approach, the level of synchronization among circadian cells was measured by the oscillation amplitudes of bioluminescence signals in individual plants. To match the amplitude information to the synchronization index on a quantitative level, a calibration was performed using a linear regression. This procedure was found simple and efficient. The advantage is that, once the calibration coefficient is derived from a single SR data set, the same value can be used for other SR data, since almost the same calibration coefficients were obtained for various cases, including two clock genes and five types of external stimuli (Supplementary Fig. 3a). This implies that for each experimental setup, the calibration is needed only once. In addition, for *PRR* mutants that show strongly suppressed oscillations, a normalization by intrinsic amplitude $A_0$ was found useful (Supplementary Fig. 3c). By this additional calibration, the PRC was estimated successfully even in the *PRR7* mutant (Fig. 2g).

Because of the simplicity and the significantly reduced experimental effort, the SR method would enable comprehensive investigation of phase responses in a diversity of organs and clock genes, which may play distinctive roles in adjusting the circadian clock to multiple environmental cues. In fact, the observed reset phases $\Theta'$ corresponded well to the timings of environmental changes in nature (i.e., most major temperature changes occur around noon and midnight). This implies that comprehensive knowledge of the SR should be useful for understanding clock-controlled activities of plants (e.g., photosynthesis, stomatal aperture, and stress responses) in nature[1,2,6]. Furthermore, the measured entrainment properties (e.g., locking phase and response strength) could be utilized for designing advanced agriculture systems, such as plant factories, intelligent greenhouses, and space farms[26,27].

Finally, this general-purpose method is applicable to other oscillator systems in science and engineering. The SR of a population of desynchronized oscillators can be easily measured

and the PRC straightforwardly constructed. The SR may elucidate entrainment mechanisms of other species or oscillators in nature and moreover can be used to control synchronized or non-synchronized state of oscillator networks.

## Methods

**Plant materials and growth conditions**. *Arabidopsis thaliana* (Columbia accession, Col-0) was used as the wild-type plant. The *CCA1::LUC* and *TOC1::LUC* reporter genes were constructed previously (the construct-c corresponding to a protein fusion[11,28]). Clock mutants (*prr5*, *prr7*, and *prr9*) with *CCA1::LUC* reporter genes were also prepared[11]. The plants were grown on 4 ml of gellan-gum-solidified Murashige and Skoog plant salt mixture with 2% (w/v) sucrose in a 40-mm dish under a 12 h light ($100\ \mu\text{mol m}^{-2}\,\text{s}^{-1}$ of fluorescent white light): 12-h dark cycle at $22 \pm 0.5\,^{\circ}\text{C}$ for 14 days. The plants were treated with 500 μl of 1-mM luciferin (made up in water) 24 h before the start of bioluminescence monitoring.

Bioluminescence measurements were carried out under illumination by light-emitting diodes of red (peak wavelength $\lambda_{\text{p}} = 660$ nm) and blue ($\lambda_{\text{p}} = 470$ nm). Supplementary Table 1 shows the light and temperature conditions for each stimulus experiment. To measure the SR, 2-h dark, 4-h/8-h dark, blue-light, 4-h $+3\,^{\circ}\text{C}$, and other temperature stimuli were applied once 262 or 274 h, 262 h, 262 h, 262 h, and 264 h, respectively, after measurement was begun. Monitoring was carried out for 14 days for all treatments of the whole plant, using an automated monitoring system (*Kondotron*) developed by Kondo et al., at measurement intervals of 20 min[29]. SR measurements for *prr7* and *prr9* were performed twice and the others were once.

In the experiments on separated shoots and roots, we separated the shoot and root on the 13th day of growth, and measured the responses against heating and cooling stimuli (4 h, ±10 °C) in the same manner as in the experiments using the whole plants.

**Data analysis**.

1) Normalization of bioluminescence: the bioluminescence was normalized as

$$\bar{X}_k = \frac{X_k - \frac{1}{2n+1}\sum_{j=0}^{2n} X_{k+j-n}}{\frac{1}{2n+1}\sum_{j=0}^{2n} X_{k+j-n}}, \qquad (2)$$

where $X_k$ is the $k$th bioluminescence signal and the window size of moving average $n$ was set to 36 for taking a moving average of the bioluminescence over 24 h.

2) Peak and trough detection: for normalized bioluminescence signal $\{\bar{X}_k: k = 1,2,\dots\}$, the slope $s_k$ at the $k$th data point was calculated as

$$s_k = \frac{1}{w}\sum_{j=1}^{w}(\bar{X}_{k+j} - \bar{X}_{k-j}) \cdot \qquad (3)$$

The moving average was set to $w = 24$. The peak point $l$ of the bioluminescence oscillation is defined as the point at which $s_l \geq 0$ and $s_{l+1} < 0$, whereas the trough point $m$ is defined as the point at which $s_m \leq 0$ and $s_{m+1} > 0$.

3) Definition of SR: denoting the stimulus end time by $t_e$, the first peak time of the bioluminescence signal after the end of stimulus by $t_p$, and the natural period by $\tau_0$, the reset phase $\Theta'$ was defined as

$$\Theta' = -\frac{t_p - t_e}{\tau_0} \times 2\pi \cdot \qquad (4)$$

Here, the free-running period $\tau_0$ was taken as the average for the first several days under constant light. The reset phase $\Theta'$ from the trough for $+10\,^{\circ}\text{C}$ stimulus in whole plant was calculated in the same manner. Since the first peak of *CCA1* appeared ~2 h after the start of the measurement, the *CCA1* peak was defined as CT2. On the other hand, the peak of *TOC1* was defined as CT14, because *TOC1* is known to have an opposite phase to *CCA1*. To compare the SRs and PRCs for different clock genes, phase $\Theta'$ was transformed to CT as $\text{CT} = \Theta'/2\pi \times 24 + 2 \bmod 24$ for *CCA1::LUC* and $\text{CT} = \Theta'/2\pi \times 24 + 14 \bmod 24$ for *TOC1::LUC*.

The reset amplitude was defined as

$$R' = \frac{\max\{\bar{X}_k\} - \min\{\bar{X}_k\}}{2}, k/3 > t_e + 12 \cdot \qquad (5)$$

The values of $\Theta'$ and $R'$ were calculated for individual plants or organs, and then their average was calculated for each, as shown in Supplementary Tables 2–4.

**Calculation of PRC using standard methods**. Method for calculating PRC by a single pulse (SP-PRC method): conditions for the growth of plants and their bioluminescence monitoring were the same as those in the SR experiment. The monitoring was carried out for 7 days. A 2-h dark pulse was applied once at 72, 78, 84, or 90 h after measurement for clock mutants (*prr5*, *prr7*, and *prr9*; Fig. 2f–h). A 2-h additional blue-light pulse once at 102, 108, 114, or 120 h (Fig. 2d). In the

analysis of the bioluminescence signals, the peak points were detected in the same manner as in the above experiments. The phase shift was defined as the difference between phases before and after the pulse, which were obtained by linear interpolation of the three (or two) peak points before and after the pulse by the least-squares method[4].

Method for calculating PRC by multiple pulses (MP-PRC method): the method for calculating PRC by multiple pulses was established in a previous study[9]. This method was used for 2-h dark (Fig. 2a); 4-h dark (Fig. 2e); and 4-h, ±10 °C (Fig. 2b, c and Supplementary Fig. 4) stimuli. Denoting the time of the $k$th peak in the bioluminescence signal by $t_p^{(k)}$ and the natural period by $\tau_0$, the phase $\Theta$ at time $t$ was defined as

$$\Theta = \frac{t - t_p^{(k)}}{\tau_0} \times 2\pi, t \in \left[ t_p^{(k)}, t_p^{(k+1)} \right). \tag{6}$$

Here, the free-running period of $\tau_0 = 23.0$ h was obtained by taking the average over $t_p^{(2)} \le t \le t_p^{(6)}$ under LL. Normalized bioluminescence was used for 4-h $-10$ °C stimulus instead of bioluminescence. Because the average time from peak to trough of the bioluminescence was 12.0 h, the phase at the trough was $\Theta = 12$ of $23 \times 2\pi$ rad.

Let the time at which each pulse is administered be $t_{stim}$. For two consecutive peaks $t_p^{(k)}$ and $t_p^{(k+1)}$ that satisfy $t_p^{(k)} \le t_{stim} \le t_p^{(k+1)}$, the phase shift $\Delta\Theta$ induced by the pulse was calculated as

$$\Delta\Theta = -\frac{t_p^{(k+1)} - t_p^{(k)} - \tau_0}{\tau_0} \times 2\pi. \tag{7}$$

The peak points that appeared between 2 h before and 4 h after the pulse were not used in the calculation, in order to exclude the ripple of data and the transient response. The phase shifts from the troughs were calculated in the same manner.

In the standard method, PRCs were extracted only from plants in which cellular oscillators are strongly synchronized with each other. Under such condition, the individual-level PRC becomes almost the same as the cellular-level PRC, making our theoretical PRC (based on a single-cell model (Eq. (8))) and the experimental PRC (measured from individual plants) comparable. According to the previous study[9], individual-level and cellular-level PRCs become non-distinguishable in experiments, when the oscillation amplitude is large enough ($A \ge 0.5$, where $A$ is defined in the previous study[9]). In the present study, we extracted only phase shift data $\Delta\Theta$ associated with such relatively large amplitude.

**Construction of PRC from SR**. Here, we introduce numerical calculation of the SR (i.e., the values of $R'$ and $\Theta'$) with respect to a pulse input based on the phase oscillator model (Eq. (1)).

First, we consider the PRC $g(\phi)$ that describes phase responses of a single clock cell. As noted in the previous subsection, this cellular-level PRC becomes the same as the individual-level PRC, when circadian cells in the individual plant are highly synchronized with each other[9]. By focusing on a single-cell oscillator, the phase oscillator model (Eq. (1)) is described as follows:

$$\frac{d\phi}{dt} = \omega + a \sin(\phi - \alpha)E(t). \tag{8}$$

Where, the external input $E(t)$ becomes 1 when the stimulus is on, and 0 when the stimulus is off. With respect to a pulse stimulus injected to the oscillator for a duration of $\Delta t$, the phase shift $\Delta\phi$ is obtained from the phase $\phi_{before}$ immediately before the pulse and the phase $\phi_{after}$ immediately after the pulse as follows:

$$\Delta\phi = \phi_{after} - \phi_{before} - \omega\Delta t. \tag{9}$$

Here, the phase variable $\phi$ in $g(\phi)$ is defined as the phase at the start of the pulse. By solving Eq. (8), we can obtain the cellular-PRCs $g(\phi)$ for various parameter values of $a$, $\alpha$, and $\Delta t$ (Supplementary Methods).

Next, from the derived $g(\phi)$, we define the SR quantities $R'$ and $\Theta'$ as follows:

$$R' e^{i\Theta'} = \frac{1}{2\pi} \int_0^{2\pi} e^{i(\phi + g(\phi) + \omega\Delta t)} d\phi. \tag{10}$$

$R'$ and $\Theta'$ are determined from the integral on the right-hand side of Eq. (10). This right-hand side shows that cellular oscillators are uniformly distributed over all phases (from 0 to $2\pi$ rad) before stimulation (i.e., singularity state), and also show that each oscillator undergoes a phase change of $g(\phi)$ by the stimulation. $R'$ and $\Theta'$ represent the order parameter (synchronization index), and the mean phase of the population of cellular oscillators after the stimulation. $R'$ is independent of $\alpha$ because, if $\alpha$ changes, only the mean phase changes in the phase distribution after a stimulus (detailed in Supplementary Methods). Thus, $R'$ is parameterized only by $a$ (and $\Delta t$) as shown in Supplementary Fig. 2a. Because $R'$ increases from 0 to 1 monotonously with $a$, $a$ is uniquely determined by $R'$. $\alpha$ is also uniquely determined by $\Theta'$, which ranges from 0 to $2\pi$ rad (Supplementary Fig. 2b). Using the estimated values for $a$ and $\alpha$, the PRC $g(\phi)$ can be recovered by Eqs. (8) and (9).

In practice, given an experimental data for SR with $R'$ and $\Theta'$, the parameters $a$ and $\alpha$ were inversely estimated step-by-step. First, by minimizing a square error between the experimental $R'$ and the theoretical $R'$ by generalized reduced gradient method, the parameter $a$ was estimated. Next, the parameter $\alpha$ was estimated to fit the theoretical $\Theta'$ to the experimental $\Theta'$. The functional relationships between $R'$-to-$a$ and $\Theta'$-to-$\alpha$ indicate that they are in a one-to-one correspondence, making

the estimation procedure straightforward without any local minima (Supplementary Fig. 2). Using these results, all experimental PRCs were estimated successfully, as shown in Fig. 2 and Supplementary Fig. 4.

**Calibration of SR**. The SR (particularly $R'$) that is determined experimentally from the signal $\{\bar{X}_k : k = 1, 2, \dots \}$ using Eq. (5) is not necessarily equal to that determined by Eq. (10). For example, if the intensity of the bioluminescence signal varies substantially among individual cells (i.e., some cells show very strong signals, and others show very weak ones), the amplitude $A(t)$ on the individual level becomes different from the synchronization index $R(t)$. Therefore, the following calibration is required:

$$R' = B(\hat{R}'), \tag{11}$$

where $R'$ and $\hat{R}'$ are the synchronization indices estimated using Eq. (10) with $g(\phi)$ and obtained by the SR experiment (Eq. (5)), respectively. If the first-order approximation (linear equation) is sufficient for $B(\hat{R}')$, then $\hat{R}' = 0$ when $R' = 0$. Therefore, it can be written simply as follows:

$$R' = \beta\hat{R}', \tag{12}$$

where $\beta$ is a calibration constant. Supplementary Figure 3 shows calibration results of the experimental data. $\beta$ was 1.61 in the experiment for various experimental stimuli and 2.06 in the experiment for mutants (Supplementary Fig. 3a, c). For the mutants, because the damping rate can vary dramatically depending on the type of mutation, we normalized $\hat{R}'$ additionally by $A_0$, i.e., the amplitude observed at the second peak of $\bar{X}_k$ after measurement has begun (Supplementary Fig. 3c). Once the calibration is carried out for one reference set of SR and PRC, no further calibration is needed for other data sets, because the same calibration constant $\beta$ can be used. On the other hand, experimental $\hat{\Theta}'$ values using Eq. (4) were in good agreement with $\Theta'$ values from Eq. (10), with $g(\phi)$ as shown in Supplementary Fig. 3b, d. Therefore, no calibration was required for $\Theta'$.

**Reporting summary**. Further information on research design is available in the Nature Research Reporting Summary linked to this article.

## Data availability

All data needed to evaluate the paper's conclusions are presented in the paper and/or Supplementary Information. Additional relevant data may be requested from the authors. Source data are provided with this paper.

## Materials availability

Additional relevant materials may be requested from the authors.

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

## Acknowledgements
We are grateful to T. Kondo for providing the bioluminescence monitoring device (Kondotron). This study was supported, in part, by Grants-in-Aid for Scientific Research (nos. 20H00423, 20H05424, and 20H05540 to H.F., and 18J20079 to K.M.) of Japan Society for the Promotion of Science and PRESTO (no. JPMJPR15O4) and CREST (no. JPMJCR15O1) of the Japan Science and Technology Agency (to H.F.).

## Author contributions
Methodology, K.M. and H.F.; resources, N.N.; investigation, K.M.; and writing—original draft, K.M., H.F., and I.T.T.

## Competing interests
The authors declare no competing interests.
