## [Peer Review File · Nature Communications]

REVIEWER COMMENTS

Reviewer #1 (Remarks to the Author):

This manuscript proposes a method to compute the phase response curve of circadian clocks. The idea is very interesting and the results look quite nice. According to my expertise, this review will focus on the (mathematical) formulation of the method, which raises several questions regarding its precision and rigor.

The method is basically composed of the following four steps:

1. Numerical computation of R' and θ' for several pairs of a and α , from equations (S7)-(S9).
The values of frequency w , Δt , and stimulus $E(t)$ are known.
This first step leads to the construction of an "inverse table", giving a correspondance between pairs (R', θ') and (a, α) .
2. Measure R' and θ' from experimental data, using equations (S3)-(S4).
3. From the table computed in step 1, obtain the "closest" pair (a^*, α^*) .
4. Simulate PRC equations (S7)-(S8) with values (a^*, α^*) to obtain the desired Phase response curve.

This sequence is essentially sound but, in my opinion, the implementation of step 1 is not very convincing and could be done using more reliable mathematical methods. The construction of an "inverse table" is a simple approach, which is necessarily very limited: how to decide which pairs (a, α) to use, how many, etc.

In addition, as it stands, the proposed method does not provide any error intervals or confidence limits.

These drawbacks in step 1 can be overcome, for instance, by first writing an optimization problem and then using a non-linear least squares type of method, gradient/Newton method, or a search method, to find the parameters that minimize a given error expression. These methods will use mathematically rigorous algorithms to find a pair (a^*, α^*) that most closely reproduces the observed pair (R', θ') .

A short, but useful overview of these methods and formulation of an optimization problem can be found in the book of Forger 2017 (see chapter 8), and references therein, for instance the book of Bradie 2006 (full references below).

Moreover, some software tools, such as the open access Scilab, have several of these methods implemented in very useful ways (see built-in functions `leastsq`, `lsqrsolve`, `fminsearch`).

The problem with using some of these gradient or search methods could be that of underdetermination: only one pair of experimental values is available for each pair of (a, α) . However, the minimum requirement seems to be met, with two known values to solve for two unknown parameters.

Another point that should be mentioned in the manuscript is the form of $E(t)$. I understand that this represents the stimulus function, but what form does it take? In particular, for the simulations in step 1, is it a constant either 1 (stimulus ON) or 0 (stimulus OFF)?

My other main comment concerns the section "Calibration of SR".

The authors say that (line 337/338) "The SR that is determined experimentally from the signal

using Eq. S4 is not always equal to that determined by Eq. S9." The authors then perform a "calibration" showing that the experimental R' relates linearly with that recovered from equation S9.

The meaning of (and the need for) this "calibration" step is a bit perplexing to me: first, the value given by S9 is the result of a parameter estimation, hence it is not necessarily "equal" to the experimental value. Second, I wonder if this "difference" between experimental and estimated value could be reduced by implementing step 1 in a more rigorous way (as suggested above). Third, it is not clear which value is used in the PRC curves to compare with data (in Figure 2, for instance).

Other small inconsistencies:

Line 133: "An increase in temperature change from $-3\text{ }^{\circ}\text{C}$ to $-10\text{ }^{\circ}\text{C}$ "  a decrease (?)

In the text (line 69) it is written 23h for frequency computation, but in Fig.3 the cycle seems to be 24h?

References:

B. Bradie 2006. A friendly introduction to numerical analysis. Upper Saddle River, NJ: Pearson Prentice Hall.

D.B. Forger 2017. Biological Clocks, Rhythms, and Oscillations - The Theory of Biological Timekeeping. Cambridge, MA: The MIT Press.

Reviewer #2 (Remarks to the Author):

The phase response curve (PRC) is a useful tool to probe resetting behavior of circadian clock systems, but PRC derivation can be time consuming and complicated and, thus, represents a roadblock to better understanding of plant circadian clock systems. To address this problem, the authors describe a rapid and simple method to derive PRCs for circadian clocks. Their approach takes advantage of what they term the Singularity Response (SR). The SR is determined by applying a resetting stimulus to a desynchronized population of individuals. The desynchronized population is prepared by extended exposure to free running conditions. Analysis of the unique resetting trajectory of each individual through modeling provides the phase and amplitude parameters needed to calculate a PRC. Modeling involves initial derivation of parameters for all possible states and then matching of experimentally-derived data to these parameters, along with calibration for all parameters. The authors compare SR-derived PRC curves to PRCs calculated with standard methods for Arabidopsis plants exposed to a range of resetting stimuli, including cooling, heating, and dark. Also, the authors calculate SR values for separated shoot and root portions of plants (referred to as different organs) for cooling and heating treatments. Comparisons of SR values obtained for these stimuli show different resetting behavior according to stimulus and organ. Taking into account entrainment evolved to re-synchronize the clock daily with day/night zeitgebers, the SR approach may represent a more biologically appropriate test of resetting compared to standard PRC derivation methods than the artificial situation of applying stimuli at different points throughout a 24 hour period.

The SR PRC idea is innovative and has merit; however, but there is a clear disparity between the conceptual description of the SR and how the SR measurement is conducted in actual practice. The authors describe SR as a consequence of "injecting a single stimulus to the cellular network in a desynchronized (i.e., singularity) state". The singularity state is described as (lines 38-42):

Singularity implies that the cellular clocks constituting the whole plant have become desynchronized from each other, diminishing the circadian amplitude of the averaged gene expressions. Because the desynchronized cellular phases are widely distributed, a single external stimulus suffices to elicit a variety of phase-dependent properties of the circadian cells simultaneously.

The singularity state is a technically accurate explanation for the behavior of weakly coupled clocks under extended free run conditions. However, the phase measurements taken here are from individual plants (representing the average behavior of many, many cells), not individual desynchronized clocks within these plants. Indeed, PRC construction requires each plant to display a discrete phase before (and after) the resetting stimulus. This inconsistency is confusing to the reader and makes interpretation of data difficult.

Fig. 1 & Fig. 2 - PRCs are helpful to understand features of circadian clock entrainment, which involves resetting to synchronize the period of the internal clock with the period of external entrainment signals. It is important to distinguish whether an SR-derived PRC reflects properties of an underlying circadian system in phase with these signals or the specific resetting behavior of the chosen reporter. An important aspect of a PRC is the reference point for $\theta = 0$ (circadian time zero, CT0). How CT0 is selected impacts the interpretation of the PRC. This commonly is "lights on" in the plant circadian clock field. What is the reference point for calculating "Phase [θ] at pulse initiation" for the diagrams in Fig 1c and Extended Data Fig. 5, or the x-axis on PRC plots such as Fig. 2? Is θ relative to a certain phase reference point derived from the reporter, the extrapolated CT0 value from prior entrainment, or another value calculated in different way? The PRCs for CCA1:LUC and TOC1:LUC shown in Extended Data Fig. 4 imply the reference point is reporter-specific, since these PRCs are mirror images of one another (compare 4a to 4c, and 4b to 4d).

In addition, if CT0 is the same for each reporter and opposite PRCs are obtained for morning-phased (CCA1:LUC) and evening-phased (TOC1:LUC) reporters, then the SR method as demonstrated appears to measure the behavior of the reporter instead of the oscillator itself. The oscillator is expected to exhibit a consistent pattern of resetting regardless of the overt behavior measured (i.e., reporter). This raises the question of what an SR-derived PRC reports and how directly comparable these PRCs are to different reporters/clock readouts, to independent experiments, and to PRCs from other methods.

Fig. 1b – Related to the issue above is the possibility the SR method examines transient behavior of the reporter instead of steady-state phase shifts of the oscillator. As shown in Fig. 1b, phase shift is measured closely following the stimulus, instead of after several cycles where the system achieves a stable phase. Have the authors ruled out this possibility? For example, by comparing phase shifts for the method shown in Fig. 1b to the phase shifts apparent after plants were allowed several cycles to reach a new steady-state phase.

Fig. 1c – This limit cycle diagram shows not all plants reach the "center of gravity" (average of all θ prime and R prime values) following the 2-h dark pulse. It is unclear whether SR-derived PRC calculation considers the θ prime and R prime values of individual plants or the average "center of gravity" of the population. What value(s) is used to produce the PRC? Also, it seems use of average values ignores all resetting behavior measured and, therefore, reduces the richness of the PRC.

Extended Data Fig. 4 c and d - A notable aspect of the temporal features of the PRCs for TOC1:LUC responding to cooling and heating is these are different from published PRCs where CT0 is pegged to the last dawn of entrainment conditions. For example, Salome and McClung (Plant Cell 2005, Figure 6H) examined resetting of TOC1:LUC with similar cooling treatments and that PRC showed phase advances in the subjective day and delays in the subjective night – opposite to the SR PRC as plotted for Extended Data Fig. 4c. A comparable opposite temporal difference exists between Extended Data Fig. 4d and a published PRC calculated for resetting of TOC1:LUC by

heating (see Thines and Harmon, PNAS 2010, Figure 3). Are these apparent discrepancies a reflection of a biological difference or the result of experimental convention?

Extended Data Fig. 5. The point of this figure is that plants in both experiments reach a similar center of gravity. However, plants in panel b show a greater diversity of trajectories than those in panel d. It appears from panel c that rhythms are quite damped and panel d indicates a more limited set of initial phases compared to panel b. A PRC derived from these data is not shown, but if one were calculated from data in panel d, will it cover a more limited array of theta prime and R prime points? The same type of plots are not shown for the PRR mutants in Extended Data Fig. 1, but the bioluminescence traces indicate severely damped rhythms for these plants. This raises the question of how do severely damped rhythms or a narrow spread of phases impact the richness of data that can be derived for constructing PRCs and is this a potential limitation to this method? Also, how difficult is it to obtain initial parameters to calculate SR and the PRC, with severely damped amplitudes just prior to the resetting treatment?

Reply

Reviewer #1:

(1) *The method is basically composed of the following four steps:*

1. Numerical computation of R' and θ' for several pairs of a and α , from equations (S7)-(S9) (new no. 8-10).

The values of frequency ω , Δt , and stimulus $E(t)$ are known. This first step leads to the construction of an "inverse table", giving a correspondance between pairs (R', θ') and (a, α) .

2. Measure R' and θ' from experimental data, using equations (S3)-(S4) (new no. 4-5).

3. From the table computed in step 1, obtain the "closest" pair (a^, α^*) .*

4. Simulate PRC equations (S7)-(S8) (new no. 8-9) with values (a^, α^*) to obtain the desired Phase response curve.*

This sequence is essentially sound but, in my opinion, the implementation of step 1 is not very convincing and could be done using more reliable mathematical methods. The construction of an "inverse table" is a simple approach, which is necessarily very limited: how to decide which pairs (a, α) to use, how many, etc. In addition, as it stands, the proposed method does not provide any error intervals or confidence limits.

These drawbacks in step 1 can be overcome, for instance, by first writing an optimization problem and then using a non-linear least squares type of method, gradient/Newton method, or a search method, to find the parameters that minimize a given error expression. These methods will use mathematically rigorous algorithms to find a pair (a^, α^*) that most closely reproduces the observed pair (R', θ') .*

A short, but useful overview of these methods and formulation of an optimization problem can be found in the book of Forger 2017 (see chapter 8), and references therein, for instance the book of Bradie 2006 (full references below). Moreover, some software tools, such as the open access Scilab, have several of these methods implemented in very useful ways (see built-in functions `leastsq`, `lsqrsolve`, `fminsearch`).

The problem with using some of these gradient or search methods could be that of underdetermination: only one pair of experimental values is available for each pair of (a, α) . However, the minimum requirement seems to be met, with two known values to solve for two unknown parameters.

The reviewer's understanding of the four steps procedure is accurate. We must admit that our original description of using the "inverse table" to estimate the parameters (α and a) did not provide our detailed procedure. In step 1, the parameters were estimated one by one: first the parameter a was estimated and then the parameter α was estimated. This is because R' is independent of α . Independence of R' on α was proven mathematically in the revised supplementary method (Eqs. (S16) and (S17)). Since R' depends only on a , its value was estimated in such a way that the model R' is fitted to the observed R' . We defined the square error between them and minimized it by the generalized reduced gradient method. Our approach is therefore based on

the optimization method, as suggested by the Reviewer. According to our numerical analysis, R' and a were found to have a simple one-to-one correspondence, making the optimization problem straightforward without any fitting error (see Supplementary Fig. 2).

Next, using the estimated a , parameter value for α was estimated so that the model Θ' was fitted to the observed Θ' . Since Θ' and α had a one-to-one correspondence, the estimation was straightforward. To demonstrate this step-by-step optimization procedure, the one-to-one correspondences of R' -to- a and Θ' -to- α are illustrated in revised Supplementary Fig. 2. Since the simple one-to-one relationship did not produce fitting errors, the confidence interval was not provided in our study.

In the revised manuscript, we clarified that our optimization has been carried out step by step, because of the parameter independence. The detailed optimization procedure was described in the main text (lines 415-428) and in the Supplementary information, where description of the "inverse table" was removed. In Supplementary information, the PRC $g(\phi)$ has been derived analytically from Eq. (8) so that R' is shown to be independent of α . We included the suggested references (Bradie 2006; Forger, 2017) as a general approach to the parameter optimization. Concerning the problem of underdetermination, it is indeed true that our one-point estimation makes the results sensitive to noise. We may overcome such weakness by fitting the model to several data sets in the future study.

- (2) *Another point that should be mentioned in the manuscript is the form of $E(t)$. I understand that this represents the stimulus function, but what form does it take? In particular, for the simulations in step 1, is it a constant either 1 (stimulus ON) or 0 (stimulus OFF)?*

$E(t)$ is 1 when the stimulus is on and 0 when the stimulus is off in Eq. (1) and Eq. (8). We have added this description in the revised manuscript (line 402-404).

- (3) *My other main comment concerns the section "Calibration of SR".*

The authors say that (line 337/338) "The SR that is determined experimentally from the signal using Eq. S4 (new no. 5) is not always equal to that determined by Eq. S9 (new no. 10)." The authors then perform a "calibration" showing that the experimental R' relates linearly with that recovered from equation S9 (new no. 10).

The meaning of (and the need for) this "calibration" step is a bit perplexing to me: first, the value given by S9 (new no. 10) is the result of a parameter estimation, hence it is not necessarily "equal" to the experimental value. Second, I wonder if this "difference" between experimental and estimated value could be reduced by implementing step 1 in a more rigorous way (as suggested above). Third, it is not clear which value is used in the PRC curves to compare with data (in Figure 2, for instance).

Mathematically, R' is defined as a synchronization index (or Kuramoto order parameter) between the cellular oscillators (Eq. (10)). Experimentally, the corresponding value \widehat{R}' is obtained from the amplitude of the bioluminescence signal, which includes information of all cellular rhythms in an individual plant. Although

the experimental \widehat{R}' certainly reflects the degree of cellular synchronization within the plant (Masuda *et al.*, 2017), it is not exactly the same as the synchronization index. Thus, calibration was indispensable to match the two indices. In addition, PRCs of individual plants can be also slightly deviated from the ideal curves due to real experimental situations (*e.g.*, time delay inevitable in the stimulus signal, slow response of the plant which broadens the sharp pulse inputs). This also requires calibration between the PRC and the synchronization indices.

In our procedure, we have calibrated the experimental \widehat{R}' first and then estimated the parameters. This procedure was found to be simple and efficient. One advantage is that, once the calibration coefficients are obtained from one SR data set, the same values can be used for other SR data. For various cases including two clock genes and five types of external stimuli, we confirmed that the calibration coefficients can be set to be the same (Supplementary Fig. 2a). Since the calibration coefficients are fixed in this way, the estimation of parameters a and α was straightforward as in our previous reply.

As suggested by the Reviewer, alternative idea is to optimize the calibration coefficient (β) together with the model parameters (a and α). However, in the present study, we did not try that possibility, because of the parameter redundancy. The calibration coefficients and the model parameters are not independent from each other in the square error function, leading to the problem of non-identifiability. Although such problem could be overcome by introducing an additional constraint, we avoided such complication in our first-step study. We would like to consider such possible extension in our future study.

We have added above explanations in the revised manuscript (lines 196-207 in Discussion).

(4) *Other small inconsistencies:*

1) *Line 133: "An increase in temperature change from $-3\text{ }^{\circ}\text{C}$ to $-10\text{ }^{\circ}\text{C}$ "  a decrease (?) R'*

Following the reviewer's comment, we have revised this sentence (line 138).

2) *In the text (line 69) it is written 23h for frequency computation, but in Fig.3 the cycle seems to be 24h?*

In Fig. 3, circadian time (CT) was used as a time-scale in order to refer the phase of circadian clock to life cycle in nature. For reader's understanding, this explanation of CT has been added in the revised manuscript (lines 94-96, 345-350).

Reviewer #2:

(1) *The SR PRC idea is innovative and has merit; however, but there is a clear disparity between the conceptual description of the SR and how the SR measurement is conducted in actual practice. The authors describe SR as a consequence of "injecting a single stimulus to the cellular network in a desynchronized (i.e., singularity) state". The singularity state is described as (lines 38-42):*

Singularity implies that the cellular clocks constituting the whole plant have become desynchronized from each other, diminishing the circadian amplitude of the averaged gene expressions. Because the desynchronized

cellular phases are widely distributed, a single external stimulus suffices to elicit a variety of phase-dependent properties of the circadian cells simultaneously.

The singularity state is a technically accurate explanation for the behavior of weakly coupled clocks under extended free run conditions. However, the phase measurements taken here are from individual plants (representing the average behavior of many, many cells), not individual desynchronized clocks within these plants. Indeed, PRC construction requires each plant to display a discrete phase before (and after) the resetting stimulus. This inconsistency is confusing to the reader and makes interpretation of data difficult.

As pointed out by the Reviewer, the SR is defined as a desynchronized state of the cellular oscillators. As a quantity to measure such state, individual cellular rhythms should be ideally measured. There exist, however, only few techniques (Muranaka *et al.*, *Sci. Adv.* 2016; Gould *et al.*, *eLife* 2018), which enable measurements of individual cellular rhythms in plant *in vivo*. Such techniques are quite costly and potentially invasive. Our aim is to develop a technique widely applicable at many laboratories. We therefore took a simple approach to measure the level of synchrony from bioluminescence signals of individual plants.

The level of synchrony is reflected on the amplitude of the bioluminescence signals because of the following reasons. In desynchronized cellular rhythms, the cellular signals cancel each other, resulting in a weak bioluminescence signal. Synchronized cellular rhythms, on the other hand, give rise to a strong bioluminescence signal, since their signals strengthen each other with coherent phases. It has been reported that the amplitude can indeed provide a good measure for the level of synchrony (Masuda *et al.*, 2017). To match the amplitude information to the synchronization index on a quantitative level, their calibration was performed (please also see our reply (3) to Reviewer #1). We have then shown that the amplitude of the bioluminescence signal provides a good measure for the level of cellular synchronization.

In the revised manuscript, we have described this issue more carefully so as not to confuse the readers.

Concerning the experimental PRCs, they were measured for individual plants by the standard method (Johnson *et al.*, 2003) as well as by the multiple pulse method (Masuda *et al.*, 2017). Thus, the measured PRCs represent phase response properties of individual plants, which show distinct phases before and after the resetting stimulus.

One of the confusions might come from the fact that the experimental PRC is measured from the individual plant, while the theoretical PRC is based on a single cell model. Our idea is that, when the cellular oscillators are highly synchronized with each other, the individual-level PRC becomes almost the same as the cellular-level PRC, making our theoretical PRC and the experimental PRC comparable. In a previous study, it has been verified experimentally that the individual-level and cellular-level PRCs become non-distinguishable, when the oscillation amplitude is large enough (Masuda *et al.*, 2017). In the calculation of the experimental PRCs (dots in Fig. (2) and in Supplementary Fig. 4), we used only phase shift data that associated with relatively large amplitude.

In the revised manuscript, we have discussed the distinction between the individual-level and cellular-level PRCs more carefully. Criterion to construct the experimental PRCs is also provided (lines 188-195 in Discussion and lines 384-392 in Methods).

(2) *Fig. 1 & Fig. 2 - PRCs are helpful to understand features of circadian clock entrainment, which involves resetting to synchronize the period of the internal clock with the period of external entrainment signals. It is important to distinguish whether an SR-derived PRC reflects properties of an underlying circadian system in phase with these signals or the specific resetting behavior of the chosen reporter. An important aspect of a PRC is the reference point for $\theta = 0$ (circadian time zero, CT0). How CT0 is selected impacts the interpretation of the PRC. This commonly is “lights on” in the plant circadian clock field. What is the reference point for calculating “Phase [θ] at pulse initiation” for the diagrams in Fig 1c and Extended Data Fig. 5, or the x-axis on PRC plots such as Fig. 2? Is θ relative to a certain phase reference point derived from the reporter, the extrapolated CT0 value from prior entrainment, or another value calculated in different way? The PRCs for CCA1:LUC and TOC1:LUC shown in Extended Data Fig. 4 imply the reference point is reporter-specific, since these PRCs are mirror images of one another (compare 4a to 4c, and 4b to 4d).*

In our theoretical model, the zero-phase (*i.e.*, $\Theta = 0$) is defined as the peak timing of the bioluminescence signal (Eqs. (4) and (6)). Since different reporter genes show different peak timings, the phase Θ depends upon the type of the reporter gene in this formula. Although our zero-phase is different from the standard CT0, which is usually defined as the time when a light signal is injected, it is straightforward to transform our phase Θ into the circadian time (CT). For instance, for CCA1:LUC and TOC1:LUC signals, their phases can be transformed as $CT = \Theta/2\pi \times 24 + 2 \bmod 24$ and $CT = \Theta/2\pi \times 24 + 14 \bmod 24$, respectively, so that their peak times become CT2 and CT14, respectively (these genes are known to show peaks at CT2 and CT14). Using the transformed CT, we have redrawn the PRCs in Fig. 2 and Supplementary Fig. 4. Although the PRCs in our previous manuscript were not consistent with the ones reported for CCA1:LUC and TOC1:LUC, the revised PRCs are in good agreement with them.

In the revised manuscript, we have added an explanation (lines 345-350).

(3) *In addition, if CT0 is the same for each reporter and opposite PRCs are obtained for morning-phased (CCA1:LUC) and evening-phased (TOC1:LUC) reporters, then the SR method as demonstrated appears to measure the behavior of the reporter instead of the oscillator itself. The oscillator is expected to exhibit a consistent pattern of resetting regardless of the overt behavior measured (*i.e.*, reporter). This raises the question of what an SR-derived PRC reports and how directly comparable these PRCs are to different reporters/clock readouts, to independent experiments, and to PRCs from other methods.*

As in our previous reply, by transforming the phase Θ' into circadian time (CT), the PRCs can be reproduced in a consistent way with the other studies. In the sense that qualitatively the same PRCs can be constructed from various reporter genes, our approach is considered to be gene-independent (although sensitivity and response properties may slightly differ from one gene to another).

(4) *Fig. 1b – Related to the issue above is the possibility the SR method examines transient behavior of the reporter*

instead of steady-state phase shifts of the oscillator. As shown in Fig. 1b, phase shift is measured closely following the stimulus, instead of after several cycles where the system achieves a stable phase. Have the authors ruled out this possibility? For example, by comparing phase shifts for the method shown in Fig. 1b to the phase shifts apparent after plants were allowed several cycles to reach a new steady-state phase.

As pointed out by the Reviewer, the SR could be affected by the transient and the results may depend upon when the phase is measured (immediately after the stimulus injection or few cycles later). We, however, consider that the transients may not have a strong influence and can be almost ignored in our study. In a previous study (Masuda et al. 2017), the PRCs subject to 2 h dark pulse were compared between two cases, in which the phase was detected immediately after the stimulus and several days after the stimulus. We found no significant difference between them (fig. S6 in Masuda et al., Sci. Adv. 2017; https://advances.sciencemag.org/content/advances/suppl/2017/10/02/3.10.e1700808.DC1/1700808_SM.pdf). Moreover, although *CCA1* and *TOC1* show anti-phase oscillations, which give rise to variability in the first peak phase (Θ') after stimulation (Supplementary Table 2), consistent PRCs were constructed from both genes (Supplementary Fig. 4). Thus, the effect of the transient seems small for determining PRC as well as SR.

(5) *Fig. 1c – This limit cycle diagram shows not all plants reach the “center of gravity” (average of all theta prime and R prime values) following the 2-h dark pulse. It is unclear whether SR-derived PRC calculation considers the theta prime and R prime values of individual plants or the average “center of gravity” of the population. What value(s) is used to produce the PRC? Also, it seems use of average values ignores all resetting behavior measured and, therefore, reduces the richness of the PRC.*

Θ' and R' were calculated for each individual plant (Eqs. (4) and (5)), and their average values were used to measure the SR. In theory, a single experiment with just one individual is enough to obtain the SR, if a complete desynchronization among cellular oscillators (i.e., mean amplitude is zero) is realized at the timing of the stimulus injection. It was, however, not very easy to realize such situation that would makes the oscillation amplitude close to zero in our experiments. As shown in Supplementary Fig. 1, a slight level of oscillation amplitude remained in each plant at the time of stimulus. These remaining amplitudes imply that the cellular oscillators are still weakly synchronized and that these coherent states, which vary from one individual to another, would result in a considerable variation in the individual singularity responses. To reduce such variation, we averaged the individual responses. In other words, the amplitude can be significantly reduced by taking an average of a population of individual plants, which is regarded as a mean data of a larger cell population. In the revised manuscript, we have added this description in Discussion (lines 188-195).

Concerning the individual information lost by this averaging, we have currently no idea to make good use of them. They are mainly due to the variability of the weakly synchronized states of the individual plants before the stimulus, which are considered as noise in our study.

(6) *Extended Data Fig. 4 c and d - A notable aspect of the temporal features of the PRCs for TOC1:LUC responding*

to cooling and heating is these are different from published PRCs where CT0 is pegged to the last dawn of entrainment conditions. For example, Salome and McClung (Plant Cell 2005, Figure 6H) examined resetting of TOC1:LUC with similar cooling treatments and that PRC showed phase advances in the subjective day and delays in the subjective night – opposite to the SR PRC as plotted for Extended Data Fig. 4c. A comparable opposite temporal difference exists between Extended Data Fig. 4d and a published PRC calculated for resetting of TOC1:LUC by heating (see Thines and Harmon, PNAS 2010, Figure 3). Are these apparent discrepancies a reflection of a biological difference or the result of experimental convention?

As in our previous reply, our model phase can be transformed into the circadian time (CT) so that the peak times of CCA1::LUC and TOC1::LUC signals can be set to CT2 and CT14, respectively. This procedure makes our model comparable to other biological experiments. In the revised Supplementary Fig. 4, consistent results were obtained with PRCs previously reported on CCA1::LUC and TOC1::LUC. Moreover, locations of the stable point observed in our PRCs agree quite well with those of the former studies, for example, around CT0 for heating (Thines and Harmon, PNAS 2010, Fig. 3) and CT12-CT18 for cooling (Salome and McClung Plant Cell 2005, Fig. 6H; Michael et al. PNAS 2003, Fig. 4).

The following changes have been made in the revised manuscript and the supplemental information as follows:

- 1) Lines 98-99: The sentence changed from “Results for another clock gene, *TIMING OF CAB EXPRESSION 1 (TOC1)*, is similar but opposite in phase (Supplementary Fig. 4).” to “Results for another clock gene, *TIMING OF CAB EXPRESSION 1 (TOC1)*, is similar (Supplementary Fig. 4).”
- 2) In Supplemental Information, new Supplementary Fig. 4 and the conversion method from Θ to CT in the figure legend have been added.
- 3) In References, Salome and McClung, Plant Cell 2005, and Thines and Harmon, PNAS 2010 have been added.

(7) *Extended Data Fig. 5. The point of this figure is that plants in both experiments reach a similar center of gravity. However, plants in panel b show a greater diversity of trajectories than those in panel d. It appears from panel c that rhythms are quite damped and panel d indicates a more limited set of initial phases compared to panel b. A PRC derived from these data is not shown, but if one were calculated from data in panel d, will it cover a more limited array of theta prime and R prime points?*

The same type of plots are not shown for the PRR mutants in Extended Data Fig. 1, but the bioluminescence traces indicate severely damped rhythms for these plants. This raises the question of how do severely damped rhythms or a narrow spread of phases impact the richness of data that can be derived for constructing PRCs and is this a potential limitation to this method? Also, how difficult is it to obtain initial parameters to calculate SR and the PRC, with severely damped amplitudes just prior to the resetting treatment?

As in our previous reply (reply (5) to Reviewer #2), such a strong damping is rather beneficial in the sense that the system can be in a strongly desynchronized state (close to singularity) at the time of the stimulus. In

Supplementary Fig. 5, data from panel c is ideal in the sense that the bioluminescence signals show strong damping and individual variation is small. On the other hand, in panel a, individual variation in oscillation amplitudes remain large at the time of the stimulus. In this case, averaging is beneficial to obtain a reliable SR. Since the center of gravity of the SR (the average values of Θ' and R') becomes the same in panels b and d, the same PRC could be constructed from both data.

As in the *PRR* mutants, individual rhythms show a strong amplitude reduction, making the SR patterns similar to those of supplementary Fig. 5d. This strong reduction of amplitude, however, affects the intensity of the SR (R'). We, therefore, calibrated this effect by normalizing R' by amplitude A_0 . By this additional calibration, the PRC was estimated successfully even for the *PRR7* mutant (Fig. 2g), which has a very strong amplitude reduction.

These explanations have been added in the revised manuscript (lines 188-195 and 204-207).

(8) Other revisions in the manuscript and the supplementary information:

- 1) In the legend of Supplementary Fig. 5, the material information “*CCAI::LUC*” was added.
- 2) In Supplementary Tables 2, 3 and 4, the values were changed slightly to correct.
- 3) Figure 2 and Supplementary Fig. 4 were redrawn using the analytical solution $g(\phi)$.
- 4) Supplementary Fig. 2 and Supplementary Fig. 4 were redrawn to be consistent with Reply for reviewers.
- 5) As supplementary methods, the analytical solution of $g(\phi)$ from Eq. (8) has been added.
- 6) Discussion and related references (red color marked) were added.

REVIEWERS' COMMENTS

Reviewer #1 (Remarks to the Author):

The authors responded to all my comments and elucidated my questions on the method for computing the PRC.

The method is now fully and clearly described in the Supplementary material. The revised version of the manuscript is also much easier to follow.

Reviewer #2 (Remarks to the Author):

I appreciate the careful responses the authors made to issues raised by my review. The revised manuscript satisfactorily addresses these issues.